# Automatic Shoreline Detection from Video Images by Combining Information from Different Methods

**Francesca Ribas** [1,*,†] **, Gonzalo Simarro** [2,†] **, Jaime Arriaga** [3,4] **and Pau Luque** [5]

1   Physics Department, Universitat Politècnica de Catalunya (UPC), Jordi Girona 1-3, 08034 Barcelona, Spain
2   Institute of Marine Sciences (CSIC), Passeig Marítim de la Barceloneta 37-49, 08003 Barcelona, Spain; simarro@icm.csic.es
3   Laboratorio de Ingeniería y Procesos Costeros, Universidad Nacional Autónoma de México (UNAM), Puerto de Abrigo, Sisal 97355, Mexico; JArriagaG@iingen.unam.mx
4   Cátedras-CONACyT, Consejo Nacional de Ciencia y Tecnología, Insurgentes Sur 1582, Ciudad de México 03940, Mexico
5   Balearic Islands Coastal Observing and Forecasting System (SOCIB), Parc Bit, 07121 Palma, Spain; pluque@socib.es
*   Correspondence: francesca.ribas@upc.edu
†   These authors contributed equally to this work.

**Abstract:** Properly registering the time evolution of the shoreline—the coastal land-water interface—is a crucial issue in coastal management, among other disciplines. Video stations have shown to be powerful low-cost tools for continuous monitoring of the coast in the last 30 years. Despite the efforts of the scientific community to get algorithms able to properly track the shoreline position from video images without human supervision, there is not yet an algorithm that can be recognized as fully satisfactory. The present work introduces a methodology to combine the results from different shoreline detection algorithms so as to obtain a smooth and very much improved result when compared to the actual shoreline. The output of the introduced methodology, which is fully automatic, includes not only the shorelines at all available times but also a measure of the quality of the obtained shoreline at each point (called self-computed error). The results from the studied beaches—located in the region of Barcelona city (Spanish Mediterranean coast)—show that such self-computed errors are in general good proxies of the actual errors. Using a certain threshold for the self-computed errors, the final computed shorelines have RMSE (Root Mean Squared Errors) that are in general smaller than 2.5 m in the great majority of analysed images, when compared to the manually digitized shorelines by three expert users. The global RMSE for all dates and beaches is of 1.8 m, with a mean bias <1 m and percentage of retrieval success >95% of the points.

**Keywords:** remote sensing; video monitoring; shoreline detection; coastal regions; weighted combination; signal filtering

## 1. Introduction

A considerable proportion of the world's sedimentary coasts are facing a general erosional trend due to, e.g., a decrease of fluvial sand supply, and this tendency will be strongly accentuated by sea level rise due to global warming [1]. In this context, monitoring the position of the shoreline—the interface between water and land—of sandy beaches is crucial to manage these dynamic environments of high socio-economic value, as well as to ensure the safety of the many cities, infrastructures and ecosystems located near them. Only if such data is available, coastal scientists can evaluate the long-term beach erosion rates or the fast morphological evolution and the inundation processes linked to storms [2,3]. Shoreline data at high spatio-temporal resolution is also needed to study beach cusps, megacusps and

shoreline sand waves, which are relevant not only due to their direct effect (the occurrence of erosional hot spots [4]) but also because they are a tool to investigate the poorly-understood sediment transport processes in the nearshore [5,6]. Measurements are essential both to perform experimental studies and to calibrate and validate morphodynamic models [7,8], an essential step before they can be used to predict the future coastal evolution under different climate scenarios. Despite the critical importance of quantifying the shoreline position, there is not yet a satisfactory solution that provides high-resolution accurate data.

Shoreline detection includes a variety of methods that range from in situ surveys to satellite images. In situ measurements are carried out nowadays with differential Global Positioning Systems (dGPS), which provide centimeter accuracy—the actual accuracy of the shoreline position is actually given by the expertise of the person carrying the dGPS receptor. While dGPS surveys are unbeatable in accuracy, they are time-expensive and its use is very limited both in the time and space domains. Remote sensing alternatives include Laser Imaging Detection And Ranging (LIDAR) [9], radar [10,11], video monitoring stations [12,13] and satellite [14–16]. In all cases, the automatic detection of the shoreline requires *ad-hoc* algorithms. The convenience of using one or another remote sensing alternative depends largely on the required continuity in time, on the space domain and on the resolution/accuracy. For instance, satellite images are likely the best choice to analyze the long term evolution of the beaches around the globe, but their spatial resolution is around 10 m in Sentinel-2—the smallest root mean squared error obtained in shoreline detection being of 5 m [15,17]—and the frequency of the measurements is low, specially in cloudy regions. Video monitoring stations, usually set to obtain hourly images and allowing to cover coastal domains of around a kilometer, are a convenient low-cost choice to study the morphodynamics of a location of interest, including the short-term processes. The resolution of the images is so that, typically, a pixel in the image represents from tens of centimeters to a few meters of the real world. Video monitoring stations started some 30 years ago (ARGUS project [13]) and typically provide three hourly images: a snapshot, a *timex* ("time exposure") image, which is typically the average of ∼600 snapshots taken at 1 Hz, and a *variance* (or *sigma*) image that includes the standard deviation, in time, of the same bunch of 600 images. The present study focuses on images form video monitoring stations, but the methodology could be applicable to satellite images as well (in fact, methods for *timex* images are in general applicable to satellite images and *vice versa*).

The literature on unsupervised shoreline detection methods—often referred to as SDMs—from video monitoring stations is extensive [18]. Although there are methods that analyze *variance* images [19–21], the majority employ *timex* images. The SDMs follow a variety of strategies based on individual pixel values to differentiate wet form dry pixels [22,23], 1D pixel information (along cross-shore transects) [24,25], or 2D pixel information (the full image) [18,26]. They also require different input, e.g., the intensity of the light [12], the red and blue channels difference [25], the full RGB information [23], the HSV color space [22], or the CIELab color space [18]. In order to improve the detection accuracy, the majority of SDMs limit their analysis to a predefined Region Of Interest (ROI). Eliminating the constraint of selecting a ROI is desirable and recent studies try to avoid its use (including, e.g., area segmentation strategies [18]). Some of the existing SDMs are fully automated but many others require a site-specific time-consuming user input to train and/or calibrate them (e.g., those using artificial neural networks [23]). Remarkably, the first proposed (1D) approach using solely the maximum intensity line of the gray-scale image (ShoreLine Intensity Maximum or SLIM model [12]) is still widely used in many studies [27], due to its simplicity and robustness, although it should be restricted to reflective environments that produce strong SLIM signals [24]. The final set of shorelines obtained from a given method in time constitutes a space-time data set that generally contains noise. Usual approaches for post-processing shorelines include space filtering [12,22].

The large amount of existing methodologies highlights the multiple challenges for an accurate, robust and universal unsupervised detection of the shoreline. Illumination and color map of the images is highly variable between sites and within the same site due to the hourly/daily inherent changes (in, e.g., sun light, rain and dust presence or beach occupation) and to the intra-annual

hydrodynamic and morphological conditions (with large changes in, e.g., wave conditions, tidal level and bar/cusps presence). Moreover, as a result of this variability, different methods work better in different dates and sites and, in fact, they may be complementary [24]. The aim of the present study is to introduce a fully-automated methodology that combines a few simple shoreline detection (raw) methods in order to obtain an accurate and robust shoreline position, giving more weight to those that are working better in a given circumstance. This philosophy to combine different independent measurements (i.e., sensor fusion), such that the resulting information has less uncertainty than that of the individual sources, is used in many fields [28,29] but it has not been tested in the framework of shoreline detection, to the authors knowledge.

The proposed methodology is applied to video-images from several beaches in the region of Barcelona city, in the Western Mediterranean Sea (Section 2.1), where manually digitized shorelines are also extracted by three expert users (Section 2.2). The four chosen raw methods are simple, based on detecting gradients in cross-shore transects in Hue (H), Saturation (S), and Value (V) channels and in the ratio Green/Red (G/R) (Section 2.3.1). Assuming that real shorelines should be relatively smooth in time and space, the weights assigned to each raw shoreline in the combination are computed, as well as the errors assigned to each measurement (Section 2.3.2). The combined shorelines are then space-time filtered using running averages (Section 2.3.3). The self-computed errors provided by the methodology also give the option to select only the points with errors smaller than a certain threshold. The obtained results for the manual shorelines, which are used as ground truth, are described in Section 3.1 and the automatic shorelines, together with the self-computed errors, are explained and compared with the manual shorelines in Section 3.2. Section 4.1 includes a sensitivity analysis to varying the parameter values of the methodology. Moreover, the results are interpreted in Section 4.2, including a set of recommendations, and the goodness of each of the four initial raw methods, based on the results analysis, is discussed in Section 4.3. Finally, the conclusions are listed in Section 5.

## 2. Methodology

### 2.1. Study Sites and Video Monitoring Stations

The proposed new methodology is presented through its application to real cases. The study sites are the beaches of Barcelona and Castelldefels, two Spanish cities located in the Western Mediterranean Sea along the Delta of the Llobregat river (Figure 1). Castelldefels city has a 4.5 km open beach with an east to west orientation [30]. It is mainly composed of light brown uniform sand with a median grain size around 0.3 mm and it presents a relatively shallow nearshore zone with a dynamic bar morphology. Barcelona city has eight man-made embayed beaches, confined between groins, with the averaged shoreline being oriented from south-east to north-west [2]. They are made of light brown sediment, with a variable grain size of some 1 mm on average, and show a steeper profile, typically without bars. In this region the tidal range can be considered negligible (<0.2 m) and waves are the main hydrodynamic forcing of the beaches [2]. The offshore mean significant wave height is 0.70 m, with maximum heights of 7.8 m, and the averaged mean period is 4.3 s. On average, calm conditions prevail during the summer period and energetic conditions occur mostly from October to May. The largest storms are those coming from the east and north-east.

Castelldefels and Barcelona beaches are video-monitored with "Argus"-like stations [13]. The five cameras in Castelldefels are atop a 30 m high tower on the promenade (Figure 1). In Barcelona, the six cameras are atop a 142 m high building in front of the marina. Both stations run currently under "Sirena" [31] and "Ulises" [32] open source software for image acquisition and processing, respectively. Castelldefels beach has been monitored since 2010, while Barcelona beaches have been monitored since 2001 (changing the set of cameras in 2015). Hereinafter, the station in Castelldefels will be referred to as CFA while the station in Barcelona will be BCN.

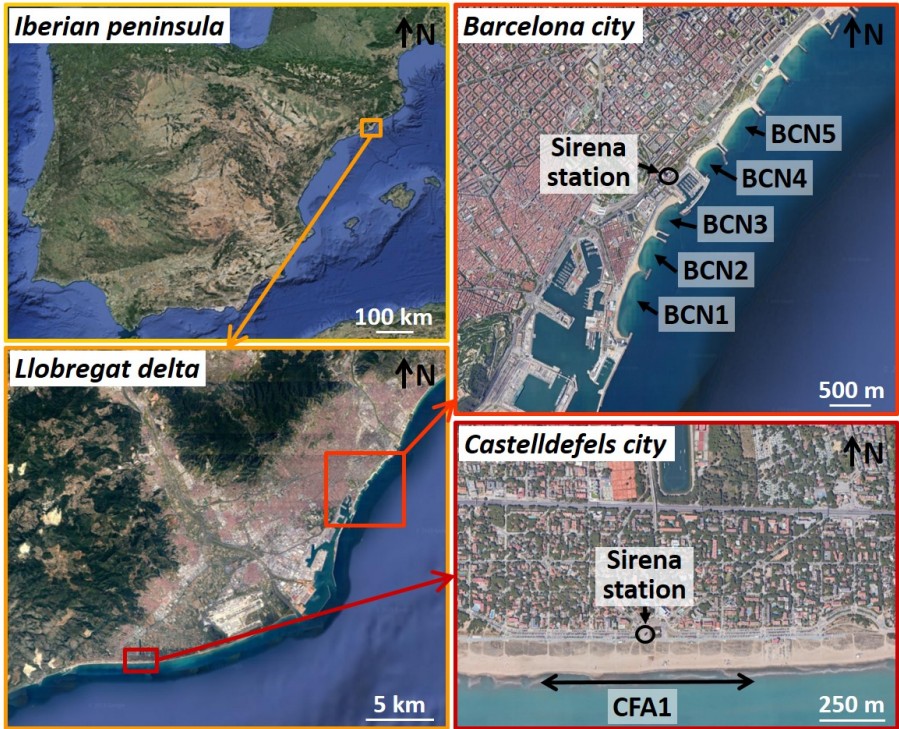

**Figure 1.** Geographical location of the studied beaches in Castelldefels and Barcelona cities (Spain) (Source: Google maps, images from TerraMetrics, CNES/Airbus, Institut Cartogràfic de Catalunya, Landsat/Copernicus, Maxar Technologies).

One snapshot, one timex and one variance image are saved every daylight hour. The calibration of the cameras allows to project the images onto a plane to obtain hourly planviews. Given that the beaches are at tideless environments, the plane is always considered to be $z = z_{\mathrm{msl}}$ (mean sea level). More than 15 Ground Control Points, as well as several points in the horizon line, are employed to perform the camera calibrations [32].

When working with the planviews, a beach is defined through two *polylines* (for land and water respectively) which are invariant in time (Figure 2, red lines). Both polylines have the same number of points, and the *transects* are defined as the lines connecting those points (Figure 2, white lines). The distance between transects is set to approximately one meter (the figure only shows a subset of transects, for clarity), and they are approximately normal to the shore, although this point is not critical for the proposed methodology.

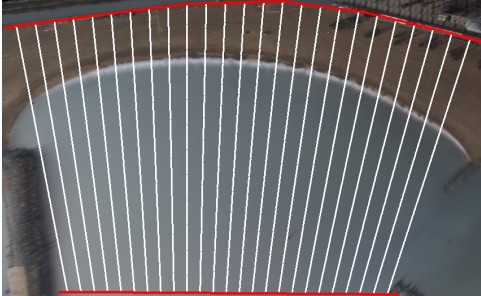

**Figure 2.** Somorrostro beach in Barcelona (beach BCN3): polylines (in red) defining the boundary and a subset of the transects (in white).

The present study focuses on an alongshore extent of 800 m of Castelldefels beach (the area closest to the video monitoring station) and on five embayed beaches of Barcelona (again, those closest to

the cameras) with alongshore widths ranging from 400 to 700 m. The studied part of the beach of Castelldefels will be referred to as CFA1. The studied beaches of Barcelona are "Sant Miquel i Sant Sebastià", "Barceloneta", "Somorrostro", "Nova Icària" and "Bogatell", but here they are referred to as BCN1 to BCN5, respectively (Figure 1). For each of the six beaches, the domain of each planview is chosen as the rectangle of minimum area that includes the whole beach (as shown in Figure 2). All the planviews are generated with 2 ppm (pixels per meter) and the size of the planviews is shown in Table 1. Of note, while the proposed methodology is applied in this work to planview images, it is also valid, with little modifications, for oblique images.

**Table 1.** Sizes in pixels (2 px = 1 m) of the planviews of each beach and corresponding number of planviews used for manual and automatic shoreline detection.

| Beach | Width [px] | Height [px] | # Manual | # Auto |
|---|---|---|---|---|
| CFA1 | 1609 | 359 | 55 | 55 |
| BCN1 | 1461 | 543 | 12 | 9 |
| BCN2 | 923 | 501 | 12 | 10 |
| BCN3 | 857 | 527 | 40 | 40 |
| BCN4 | 749 | 603 | 12 | 12 |
| BCN5 | 1293 | 743 | 12 | 12 |
| total | | | 143 | 138 |

### 2.2. Manual Shoreline Digitization

The ground truth to test the proposed automatic method is a set of 143 shorelines picked manually from the corresponding planviews. For each of the 143 planviews, three expert users have selected a set of points along the visually detected shoreline (see examples in Figures 3A, 4A and 5A). The error associated to the manual digitization will be later on assessed by comparing the differences in the shorelines obtained by the three users.

The choice of the 143 planviews is meant to study the intra-year and intra-day variability (changes in sun height, wave conditions and amount of beach users). To analyze the intra-year variability, one image has been selected for each month (noon of day 15th) in 2015 for the beach of CFA and in 2017 for the five beaches of BCN. To take into account the intra-day variability, several extra images per day have been taken in 18 different days of beaches CFA1 and BCN3. These days have been mainly selected in winter time, in order to test the method capability to cope with the tougher conditions given by more energetic waves and a lower sun height. In total, 55 planviews are being studied of beach CFA1, 12 planviews of beaches BCN1, BCN2, BCN4 and BCN5 and 40 planviews of beach BCN3 (Table 1).

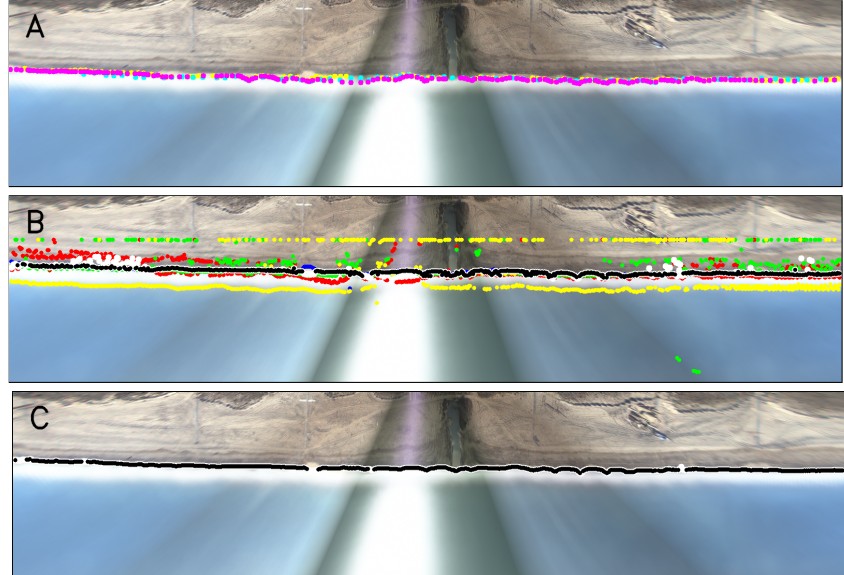

**Figure 3.** Examples of (**A**) manual shorelines digitized by the three users (in different colors), (**B**) raw shorelines out of the four methods (red, green, blue and yellow) and combined shoreline (white and black) and (**C**) filtered shoreline (white and black) of beach CFA1 on 27 November 2017 at 12 h. In (**B**), the raw shorelines come from Hue gradient (red), Saturation gradient (green), Value gradient (blue) and R/G gradient (yellow). In (**B**), the white points are the combined shoreline, while the points of the combined shoreline where $E^c < 10$ px are in black (Equation (5)). In (**C**), the white points are the filtered shoreline, while the points of the filtered shoreline where $E^f < 10$ px are in black (Equation (9)). Sun glare occurs in the central camera and some areas also show beach cusp presence.

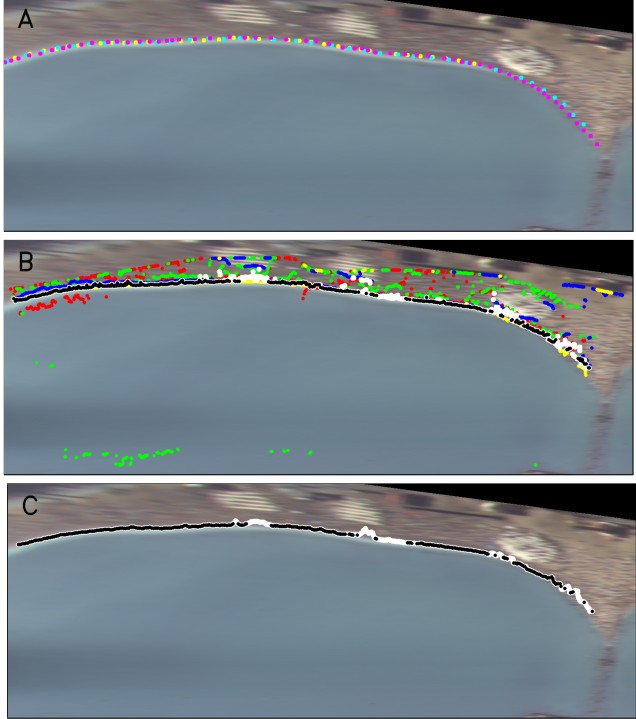

**Figure 4.** Equivalent to Figure 3, but of beach BCN1 on 15 July 2017 at 12 h. This planview shows an example of high beach occupation.

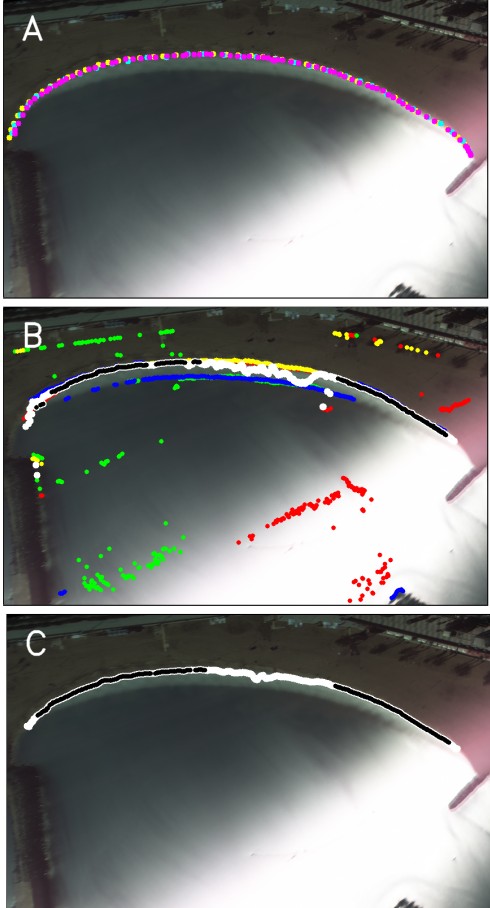

**Figure 5.** Equivalent to Figure 3, but of beach BCN3 on 7 November 2017 at 11 h. There is a strong contrast between the area with sun glare and the rest of the image, a phenomenon that occurs often in the studied planviews.

### 2.3. Automatic Shoreline Detection

Four different methods are used in this work to find the first guesses of the shoreline. These guesses, called hereafter "raw" shorelines, are noisy and may have large errors when compared to the ground truth shorelines. The idea of this work is to do a weighted combination of these raw shorelines to obtain a better approximation of the shoreline. The basic hypothesis to define the weights is that the smoother a raw shoreline behaves, both in space and time, the better it is and the more weight will have at a given space-time position. It is critical, therefore, that the raw methods provide unfiltered shorelines. While in this work four particular methods are considered, the extension to an arbitrary number of other raw methods is straightforward. Once the raw shorelines have been combined, the resulting combination is space-time filtered. The three steps (raw shoreline acquisition, combination and filtering) are described in detail below.

### 2.3.1. Raw Shorelines

The four methods used to obtain the raw shorelines are inspired by well known methods in the literature. They are chosen to be particularly simple, but still giving good estimates of the shorelines under some circumstances. The four methods are equivalent except for the fact that they use different channels of the planview image: Hue, Saturation, Value and the ratio Red/Green (which is not actually a channel but a ratio of two RGB-channels). For each of the four channels the procedure is as follows. In every cross-shore transect (Figure 2), the distribution of the channel value is first smoothed using a

Butterworth filter with a characteristic length of 15 m. Then, the gradient of the smoothed channel is considered to obtain a proxy of the shoreline position.

Some authors have used jointly Hue and Saturation to discriminate sand and water [22]. Hue represents the color, where the blueness of the water typically has higher values than the yellowish/brownish color of the analyzed beaches (e.g., water is lighter in Figure 6A, which means larger Hue values). Thereby, for this channel we consider the shoreline position of each transect as the maximum value of the gradient from land to water. The other channel that complements the color definition is Saturation, which describes the color purity. Since the water zone has often slightly lower Saturation values in the studied beaches (e.g., darker in Figure 6B), the shoreline is taken as the maximum value of the gradient from water to land. The Value channel in the HSV space is equivalent to the Intensity of the grayscale images used by the SLIM algorithm [12]. This channel is very useful to detect the shoreline when there is sun glare. A land-to-water increase is expected when the sun is reflected by the water, which occurs quite often in the studied beaches (e.g., Figure 3, middle part). However, a land-to-water decrease can also occur because more light can be reflected to the camera by the sand than by the water under calm conditions if the sun is behind the camera. For this reason, we consider the maximum of the absolute value of the gradient of this channel to obtain the shoreline. The last method, using the ratio Red/Green, is inspired by other works in the literature (e.g., [33]). Usually, this ratio is higher in the land than in the water (e.g., Figure 6D) because the sand tends to reflect more the red and the water tends to reflect more the green. However, the opposite situation has also been observed under certain light conditions and, therefore, as for the Value channel, we consider the absolute value of the gradient of the ratio to obtain the shoreline. The four methods are very simple and produce a noisy behaviour, and the present methodology focuses on how to combine them to obtain an accurate shoreline.

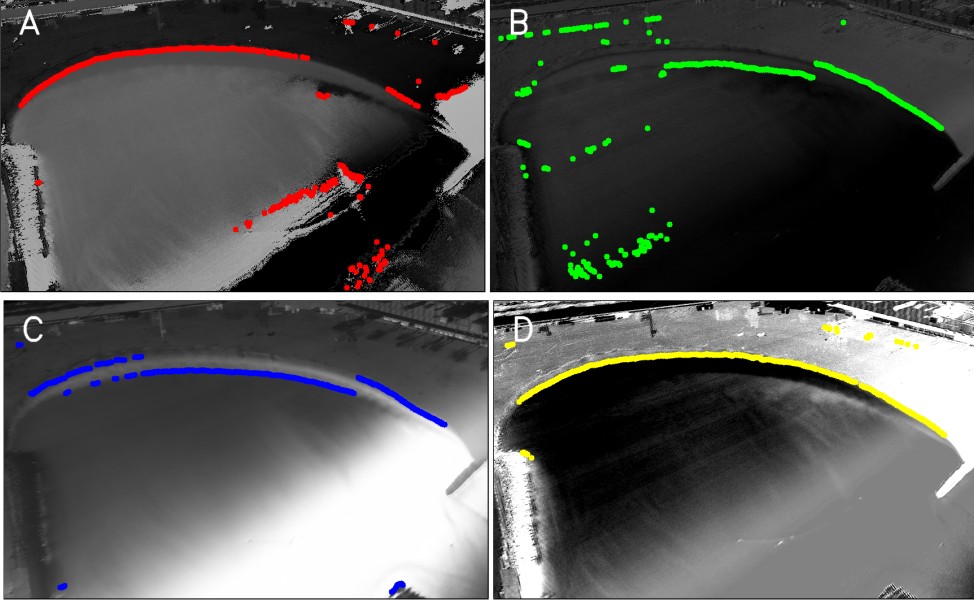

**Figure 6.** Examples of (**A**) Hue channel, (**B**) Saturation channel, (**C**) Value channel and (**D**) Red over Green channels, with the corresponding raw shorelines, of beach BCN3 on 7 November 2017 at 11 h (same planview as in Figure 5).

For all beaches, the raw shorelines are obtained every daylight hour during a period that encompasses, with a margin of one month, the dates of the manual shorelines. The raw shorelines are not computed if the planviews are too dark, namely when the average Value of the HSV-planview is below 90 (Value ranging in $[0, 255]$). Note that 5 out of 143 manually treated planviews are disregarded for being too dark: on 15 January 2017 and 15 February 2017 of beaches BCN1 and BCN2 and on 15 December 2017 for BCN1.

### 2.3.2. Weighted Combination of the Raw Shorelines

To perform the weighted combination of the raw shorelines for a specific date and transect, the weights must be first computed (they may vary from transect to transect for one date). To find them, the first step is to compute a space and time filtered version of each raw shoreline, since the weights of each raw shoreline are to be inversely proportional to the distances to the corresponding filtered version.

We first take a time interval around the desired date of $\Delta\tau$ days to build a set of (four) matrices $D^k$ (with $k$ standing for the raw method). Each matrix has as many columns as dates around the given date and as many rows as transects in the corresponding beach. The values of the matrices are the shoreline positions (the distance to the landward limit of the beach through the transect) of the raw method for each transect and date. Each of the $k$ matrices is first clipped and smoothed in time in the following way: for each transect $i$, the time history of the shoreline position, $D_{ij}^k$, which is a function of time $j$ only, is fitted through a polynomial of order $a_t$. Being $e_{t,i}$ the Root Mean Squared Error (RMSE hereafter) of $D_{ij}^k$ relative to the polynomial fitting, the shoreline position is clipped (limited) to the fitted value $\pm e_{t,i}$. A Butterworth filter with characteristic length $\Delta t$ is applied to the clipped signal (Figure 7). The parameters $a_t$ and $\Delta t$, together with $\Delta\tau$, are user defined. The proposed default values are introduced in Table 2 (case 01). Likewise, it follows a spatial clipping and smoothing of the already time-wise filtered matrix. Now, the parameters are $a_s$ and $\Delta s$ (Table 2). Flipping the order of the smoothing procedure (i.e., first in space and then in time) did not appreciably affect the results.

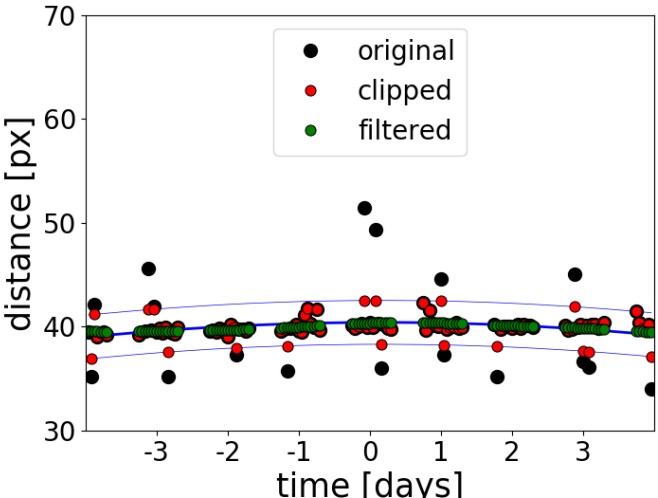

**Figure 7.** Illustration of time-wise clipping and filtering of matrix $D_{ij}^k$. Black dots stand for the original $D_{ij}^k$ values, as a function of time ($j$); blue lines stand for the polynomial fitting and the limits for clipping (polynomial $\pm e_{t,i}$); red dots are clipped data; green dots are filtered data. In this illustrative example, $\Delta\tau = 4$ days, $a_t = 2$ and $\Delta t = 2$ days.

Being $\mathcal{D}_{ij}^k$ the result of filtering the matrix $D_{ij}^k$ in space and time, the weight for each raw method, transect and date, $w_{ij}^k$, is here assumed to be inversely proportional to the distance between the two shorelines—unfiltered and filtered, i.e., $D_{ij}^k - \mathcal{D}_{ij}^k$. This is a proxy of the "noise" of the signal so that the smoother the signal, the larger the weight. We use

$$w_{ij}^k = \frac{1}{\left(|D_{ij}^k - \mathcal{D}_{ij}^k| + \varepsilon_1\right)^{\alpha_1}}, \tag{1}$$

where $\varepsilon_1 \ll 1$ is included to avoid infinite (here, $\varepsilon_1 = 10^{-6}$ px). The default value of $\alpha_1 = 2$ (Table 2) implies that the weight is the inverse of a variance, a standard procedure to combine estimators from different sources [28]. Finally, the combined shoreline matrix is the weighted combination of the four raw matrices of shorelines, i.e.,

$$\mathrm{D}_{ij}^c = \frac{\sum_k w_{ij}^k \mathrm{D}_{ij}^k}{\sum_k w_{ij}^k}. \tag{2}$$

Note that the combined shorelines in $\mathrm{D}_{ij}^c$ use the unfiltered raw signals; the filtering process has been employed, so far, only to obtain the weights for the combination. Figures 3B, 4B and 5B show examples with the points in white corresponding to the combined shorelines.

**Table 2.** Values considered for the combination and the filtering steps, where – indicates that it is the same value as in case 01 (default).

| Case | Time | | | Space | | | | Filtering | | |
|------|------|------|------|------|------|------|------|------|------|------|
| | $\Delta\tau$ [days] | $a_t$ [-] | $\Delta t$ [Days] | $a_s$ [-] | $\Delta s$ [px] | $\alpha_1$ [-] | $b_s$ [-] | $\alpha_2$ [-] | $m_t$ [-] | $m_s$ [-] |
| 01 | 15 | 2 | 0.250 | 4 | 20.0 | 2.0 | 4 | 1.0 | 3 | 5 |
| 02 | – | – | 0.125 | – | – | – | – | – | – | – |
| 03 | – | – | 0.500 | – | – | – | – | – | – | – |
| 04 | – | 1 | – | – | – | – | – | – | – | – |
| 05 | – | – | – | 2 | – | – | – | – | – | – |
| 06 | – | – | – | – | – | 1.0 | – | – | – | – |
| 07 | – | – | – | – | – | – | 2 | – | – | – |
| 08 | – | – | – | – | – | – | – | 0.5 | – | – |
| 09 | – | – | – | – | – | – | – | 2.0 | – | – |
| 10 | – | – | – | – | – | – | – | – | 1 | – |
| 11 | – | – | – | – | – | – | – | – | 5 | – |
| 12 | – | – | – | – | – | – | – | – | – | 3 |
| 13 | – | – | – | – | – | – | – | – | – | 7 |

To keep track of the quality of the combined shorelines for each transect and time, we define

$$\mathrm{E}_{ij}^{c0} = \sqrt{\frac{\sum_k w_{ij}^k \left(\mathrm{D}_{ij}^k - \mathrm{D}_{ij}^c\right)^2}{\sum_k w_{ij}^k}}, \tag{3}$$

which is a weighted RMS (Root Mean Squared) distance of the shoreline signals compared to the combined one. A second quantity $\mathrm{E}_{ij}^{c1}$ is defined as

$$\mathrm{E}_{ij}^{c1} = |\mathrm{D}_{ij}^c - \mathcal{D}_{ij}^c|, \tag{4}$$

where $\mathcal{D}_{ij}^c$ is the result of fitting (space-wise) each combined shoreline through a polynomial of order $b_s$ (Table 2). Whilst $\mathrm{E}_{ij}^{c0}$ measures the scattering of the best raw signals compared to the combined signal, $\mathrm{E}_{ij}^{c1}$ gives a measure of the spatial noise of the combined signal. We finally define the "self-computed" error as

$$\mathrm{E}_{ij}^c = \mathrm{E}_{ij}^{c0} + \mathrm{E}_{ij}^{c1}, \tag{5}$$

which measures the self-assessed quality of the combined shoreline position for a given transect and date (recall that $i$ stands for the transect and $j$ for the date). The points of the combined shoreline where $\mathrm{E}_{ij}^c$ is smaller than a certain threshold will be considered as "good" points (e.g., see the black dots in Figures 3B, 4B and 5B for $\mathrm{E}^c < 10$ px).

### 2.3.3. Filtering of the Combined Shorelines

To obtain the final shorelines, we apply two moving weighted-average filters (in time and in space) to the combined shorelines $D_{ij}^c$, using a power of the inverse of the errors $E_{ij}^c$ as weights (similar to Equation (1)),

$$w_{ij}^c = \frac{1}{\left(E_{ij}^c + \varepsilon_2\right)^{\alpha_2}},\tag{6}$$

where $\varepsilon_2 = 10^{-6}$ px and the value of $\alpha_2$ is given in Table 2. This table also displays the windows sizes (in number of points) used for the moving averages in time and space, called $m_t$ and $m_s$, respectively. The application of the filtering in time gives

$$D_{ij}^{f0} = \frac{\sum_l w_{il}^c D_{il}^c}{\sum_l w_{il}^c},\tag{7}$$

with $l$ including the closest $m_t$ points in time. The spatial filter is finally applied to get

$$D_{ij}^f = \frac{\sum_l w_{lj}^c D_{lj}^{f0}}{\sum_l w_{lj}^c},\tag{8}$$

with $l$ including the $m_s$ points nearest to the point of interest.

A measure of the self-computed error for the filtered result is considered, similarly, as an average of the weighted combination of the errors $E^c$ (in space and time). Namely,

$$E_{ij}^f = \frac{1}{2}\left[\frac{\sum_l w_{il}^c E_{il}^c}{\sum_l w_{il}^c} + \frac{\sum_l w_{lj}^c E_{lj}^c}{\sum_l w_{lj}^c}\right].\tag{9}$$

Figures 3C, 4C and 5C show examples with the points in white corresponding to the filtered shorelines and the points in black to the filtered shorelines for which $E^f < 10$ px.

## 3. Results

The manual shorelines, which are considered as the ground truth to test the automatic method, are first shown and analyzed. Their inner error, computed as the RMS distances between the points digitized by the different users, is quantified. The actual error associated to one automatic shoreline (combined or filtered) is then computed as the RMS of the distances of the manual points to the automatic shoreline. The actual errors are subsequently compared to the self-computed errors. Finally, the RMSE of the combined and filtered shorelines are analysed.

### 3.1. Manual Shorelines

Examples of the manually digitized shorelines by the three expert users have already been shown in Figures 3A, 4A and 5A. In order to evaluate the experimental errors of the ground truth data the differences between users are first computed for each beach and date. First, we compute the distances (in pixels) from each point digitized by a user "X" to the polyline passing through all the points of another user "Y". The RMS of the distances corresponding to a complete image is used to estimate the error of the whole manual shorelines digitized by these two users in that image. Figure 8 displays these RMS distances in pixels between users, i.e., the RMSE of the manual shorelines, in the different dates for beaches CFA1, BCN1 and BCN3. Finally, the global RMSE in pixels of the manual shorelines in each beach are computed as the mean for all dates and users of the RMS distances between users (last column in Table 3).

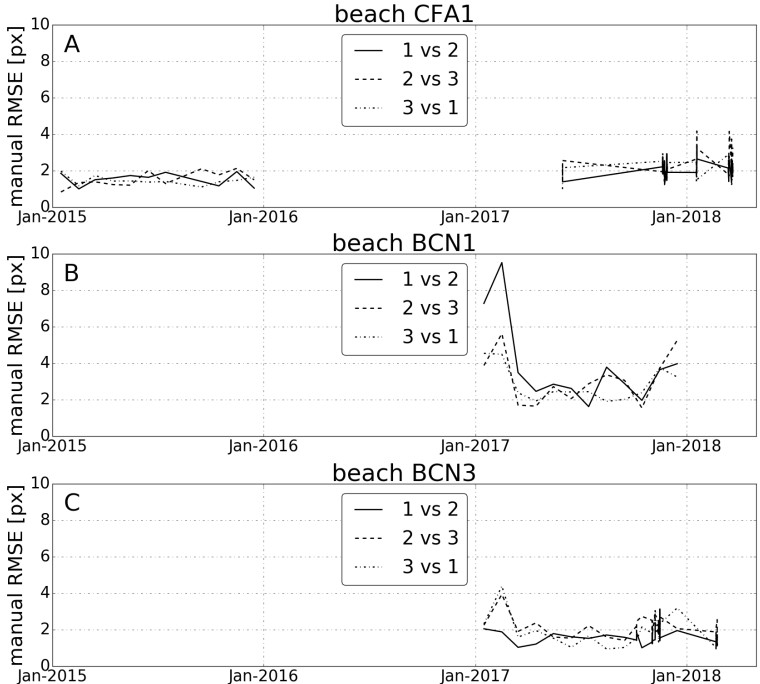

**Figure 8.** Root Mean Square Error (RMSE)—RMS distance between users—in pixels (2 px = 1 m) between the manual shorelines digitized by the three expert users in the different dates of beaches (**A**) CFA1, (**B**) BCN1 and (**C**) BCN3.

**Table 3.** Global actual Root Mean Squared Error (RMSE) and bias (in meters and including all the studied dates), together with success percentage of the combined shorelines and the filtered shorelines, using no threshold for the self-computed errors and with thresholds of 20 and 10 px (the same threshold is applied to the combined and the filtered shorelines). Bold numbers indicate the recommended configuration. The RMSE of the manually digitized shorelines for each beach is also given in the last column.

| Beach | Threshold [px] | Combined | | | Filtered | | | Manual |
|---|---|---|---|---|---|---|---|---|
| | | RMSE [m] | Bias [m] | Success [%] | RMSE [m] | Bias [m] | Success [%] | RMSE [m] |
| CFA1 | none | 2.5 | 0.7 | 100 | 2.2 | 0.9 | 100 | |
| | 20 | 1.9 | 0.8 | 95 | **2.0** | **0.9** | **96** | 1.1 |
| | 10 | 1.6 | 0.8 | 73 | 1.7 | 1.0 | 76 | |
| BCN1 | none | 2.7 | 0.2 | 100 | 2.3 | 0.3 | 100 | |
| | 20 | 2.3 | 0.4 | 94 | **2.1** | **0.5** | **95** | 1.4 |
| | 10 | 2.0 | 0.5 | 61 | 1.9 | 0.4 | 79 | |
| BCN2 | none | 5.2 | −1.4 | 100 | 2.3 | −1.2 | 100 | |
| | 20 | 1.6 | −0.8 | 95 | **1.7** | **−1.1** | **97** | 1.3 |
| | 10 | 1.6 | −0.8 | 78 | 1.6 | −1.1 | 84 | |
| BCN3 | none | 1.6 | −0.3 | 100 | 1.0 | −0.2 | 100 | |
| | 20 | 1.3 | −0.2 | 98 | **1.0** | **−0.2** | **99** | 1.0 |
| | 10 | 1.0 | −0.1 | 90 | 1.0 | −0.1 | 93 | |
| BCN4 | none | 2.3 | −2.0 | 100 | 2.5 | −2.3 | 100 | |
| | 20 | 2.3 | −2.0 | 100 | **2.5** | **−2.3** | **100** | 0.7 |
| | 10 | 2.3 | −2.0 | 99 | 2.5 | −2.3 | 100 | |
| BCN5 | none | 3.9 | −0.1 | 100 | 2.9 | −0.4 | 100 | |
| | 20 | 1.6 | −0.8 | 93 | **1.5** | **−0.9** | **94** | 0.8 |
| | 10 | 1.6 | −0.9 | 77 | 1.6 | −1.0 | 81 | |

The three expert users obtain shorelines that are in general very similar in the majority of analyzed beaches and dates. For beaches CFA1, BCN3, BCN4 and BCN5 the global error for all users and dates are of about 2 px, which translates to 1 m in the real world (see Figure 8A,C and Table 3). The discrepancies between users are slightly higher in BCN1 and BCN2, of about 3 px (Figure 8B). Particularly high errors occur on 15 January 2017, with distances up to 7 px in both beaches, and on 15 February 2017, with distances up to 9 px in BCN1. The reason is that the corresponding original planview images were much darker than usual, which makes the visual shoreline detection especially difficult. In fact, the corresponding raw shorelines are not extracted for being too dark (together with 15 December 2017 of beach BCN1). This is automatically decided in one of the steps to extract raw shorelines, as explained in Section 2.3.1. Thereby the corresponding manual shorelines are not used and the global error in these beaches has been computed excluding these dates. The global RMSE of the manual shorelines for each beach is given, in meters, in the last column of Table 3.

### 3.2. Automatic Shorelines

The combined and filtered shorelines have been extracted for all the studied dates and beaches using the default parameter values corresponding to case 01 in Table 2. Examples are shown in Figures 3B,C, 4B,C and 5B,C. To evaluate the actual errors of these automatic shorelines, with respect to the manual shorelines, two statistical descriptors are used, namely the bias (or averaged error) and the RMSE. The bias represents a systematic error of the automatic shorelines, with a positive bias indicating here that the automatic shoreline is shoreward displaced with respect to the manual shoreline (ground truth). The RMSE includes the systematic errors and the dispersion (random) errors, and gives more weight to the largest errors. The transects that are too close to a groin (e.g., southern groin of beach BCN3, see Figure 5) are not considered for the comparison between manual and automatic shorelines. Also, given that the presented methodology uses information of the nearby spatial points, the extreme points of the automatic shorelines show larger errors (see Figure 5C). Thereby, the four extreme transects of each side of each beach are excluded from the analysis.

For each point of the manual shorelines, the first step is to compute the distance to a line connecting the two closest points of the automatic shoreline (combined or filtered). This distance is computed with a sign, so that the sign changes depending on which side of the line the point is. The corresponding self-computed error ($E_{ij}^c$ and $E_{ij}^f$) for that point is quantified as a weighted average of the self errors of the two closest automatically obtained points. The RMSE and bias of the whole automatic shoreline in an image is then computed as the RMS and the average, respectively, of the distances (with signs) to the manual points in that image. The computation of the actual errors in an image can be performed for all the points, i.e., without applying any threshold to the self-computed errors. However, as explained before, it can be useful to automatically exclude from the analysis the points that have an associated self-computed error larger than a certain threshold. In that case, the percentage of success, i.e., the percentage of manual points for which the corresponding automatic shoreline point presents a self-computed error below the threshold, is also computed. Finally, the actual RMSE, bias and success percentage for each beach including all dates are also evaluated both without applying a threshold in the self-computed errors and for different threshold values ranging from 5 to 40 px.

Before comparing the automatic and the manual shorelines by analysing the obtained errors, the usefulness of the self-computed errors is evaluated by comparing them with the actual RMSE obtained considering all the points. As shown in Figure 9A–C, the self-computed errors are a factor 2 larger than the actual errors but there is a significant correlation between the two quantities. For the case of the self-computed errors of the combined shorelines, the Pearson correlation coefficient is larger than 0.75 for all the studied beaches, except beach BCN4, where the correlation is not significant ($r < 0.4$). This beach shows a singular general behaviour discussed later on in this section. In the case of the filtered shorelines, the values of the correlation coefficients between the self-computed and the actual errors are lower but the correlation is still significant ($r > 0.55$) except, again, in beach BCN4 ($r < 0.4$). Thereby, the self-computed errors are better proxies of the actual errors in the

combined shorelines than in the filtered shorelines. When the correlation in the filtered shorelines is not high, the reason is usually that there are self-computed errors that are too large compared with the corresponding actual errors (an example can be seen in Figure 9C, rounded points down at the center). The contrary, i.e., outliers above the trend of the cloud, do not usually appear and therefore the self-error of the filtered shoreline turns out to be a conservative estimate of the actual error.

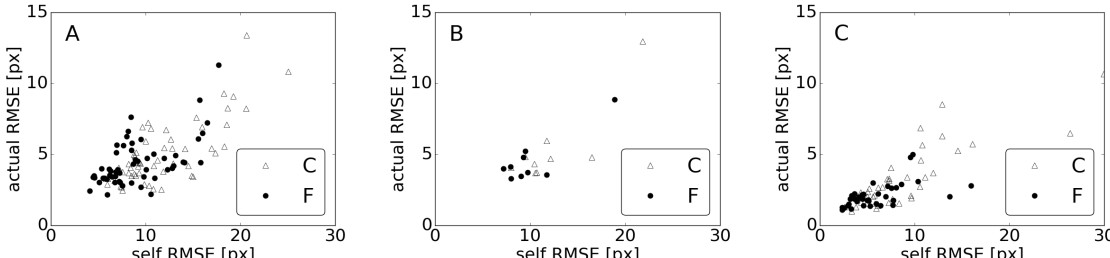

**Figure 9.** Actual RMSE—RMS distance from manual to automatic shorelines—versus self-computed error, in pixels, for all planviews of beaches (**A**) CFA1, (**B**) BCN1 and (**C**) BCN3, both for the combined (empty triangles) and the filtered (filled circles) shorelines.

Once the validity of the self-computed errors has been assessed, the dependence of the actual errors on using a certain value for the threshold in the self errors is studied, taking also into account the percentage of success. When decreasing the value of the threshold (i.e., being more restrictive), both the actual RMSE and the success percentage decrease for both the combined and the filtered shorelines (see examples in Figure 10A–C). In the rest of the section, we discuss the results for two threshold values: 20 px since it decreases the actual RMSE without significantly decreasing the percentage of success, and 10 px as an example of more reliable computation but with more gaps. The results without threshold (all points are kept) are also given for comparison.

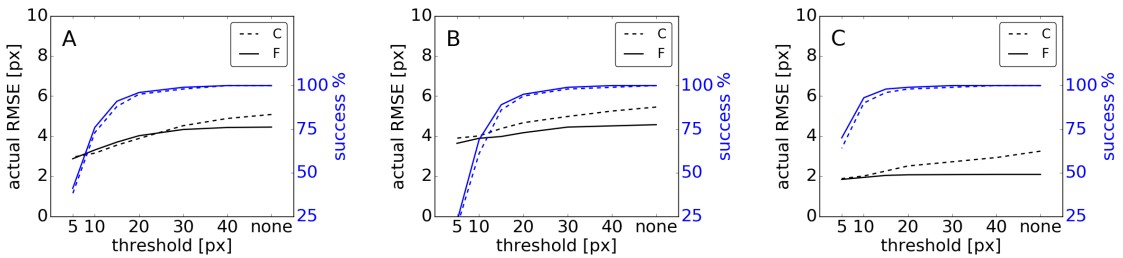

**Figure 10.** Actual RMSE (RMS distances in pixels between manual and automatic shorelines, in black) and success percentage (in blue) for all dates using different thresholds for the self-computed errors of beaches (**A**) CFA1, (**B**) BCN1 and (**C**) BCN3, for the combined (dashed) and the filtered (solid) shorelines.

The automatic shorelines are in general good representations of the manual shorelines, as shown in Table 3 (with the errors given in meters, with 1 m = 2 px, to facilitate interpretation). The actual RMSE of the combined shorelines for all dates without applying any threshold are already limited (<3 m) in all beaches, except in BCN2 (∼5 m) and BCN5 (∼4 m). These errors strongly diminish (<2.5 m in all beaches) when applying the default threshold value of 20 px while the percentages of success remain high (⩾93%). In case of a threshold of 10 px the errors decrease again in three of the beaches but the success percentage diminish significantly in all of them (up to some 60%). Filtering the combined shoreline, even without using a threshold, is another method that also significantly diminishes the actual errors, which become <3 m in all beaches (Table 3, filtered shorelines). The actual errors of the filtered shorelines also diminish (⩽2.5 m in all beaches) when applying the threshold value of 20 px, maintaining large percentages of success (⩾94%). Using a threshold value of 10 px only diminishes the error in two beaches but paying the price of a large decrease in percentage of success in all of them.

The examples shown in Figures 3C, 4C and 5C of filtered shorelines (with a threshold of 10 px) have final RMSE of 2.0 m with 93% success in CFA1, 1.7 m with 76% success in BCN1 and 1.1 m with 66% success in BCN3. When increasing the threshold to 20 px, many more points are taken so that the success percentages increase to more than 95% in the three examples and the errors only increase to 2.4 m in BCN1 and 1.5 m in BCN3 (it remains equal in CFA1).

Beach BCN4 shows again a peculiar behaviour, the thresholds playing no role at all. This behaviour is related with the bias, which shows a different pattern depending on the beach (Table 3). Beach CFA1 shows a slightly positive bias, i.e., a small onshore displacement of the automatic shorelines relative to the manual shorelines. Beaches BCN1 and BCN3 show hardly no bias and beaches BCN2 and BCN5 show a small negative bias. The most outstanding case is that of beach BCN4, which shows a relatively large negative bias of the order of 2 m. In fact, the actual RMSE in that beach is mainly related with the bias, i.e., due to a systematic offshore displacement of the automatic shorelines with respect to the manual shorelines. This explains why the self-computed errors, which are small in this beach (not shown), do not correlate with the actual errors ($r < 0.4$), and also why reducing the threshold does not have an effect on the actual errors in that beach (Table 3).

Figure 11A–C shows the actual RMSE obtained for the different planviews of beach CFA1 (55 dates) and beaches BCN1 and BCN3 (9 and 40 dates respectively) for both the combined and the filtered shorelines, and both using a threshold value of 20 px and without any threshold. In BCN1 the decrease of the large error in the combined shoreline of the last planview is obtained when applying both the filtering and the threshold (Figure 11B), and the same occurs in BCN2 (not shown). In BCN3, the combined shorelines have large errors in a few dates that diminish applying either the threshold or the filtering, as shown in Figures 5 and 11C. BCN4 beach shows a different behaviour, with 3 (out of the 12) shorelines still showing errors up to 7 px (3.5 m) in the combined shorelines that do not improve neither applying the threshold nor the filtering because they are related to large biases. Finally, in BCN5 the decrease of the largest errors is associated with the application of the threshold (and not with the filtering process). This also occurs in CFA1, but a few planviews still show errors $> 6$ px even after applying the threshold, as shown in Figure 11A. In particular, this occurs for 5 (out of 55) planviews if the threshold is 20 px but it would be reduced to only 1 case using a threshold of 10 px. In any case, even using the less restrictive threshold of 20 px the actual errors for the filtered shorelines are $<5$ px ($<2.5$ m) for the large majority of analysed planviews (117 out of 138). The mean of the RMSE, bias and success percentage using a threshold of 20 px including all dates and beaches are 1.7 m, $-0.03$ m and 96% for the combined shorelines and 1.7 m, $-0.02$ m and 97% for the filtered shorelines (first line in Table 4).

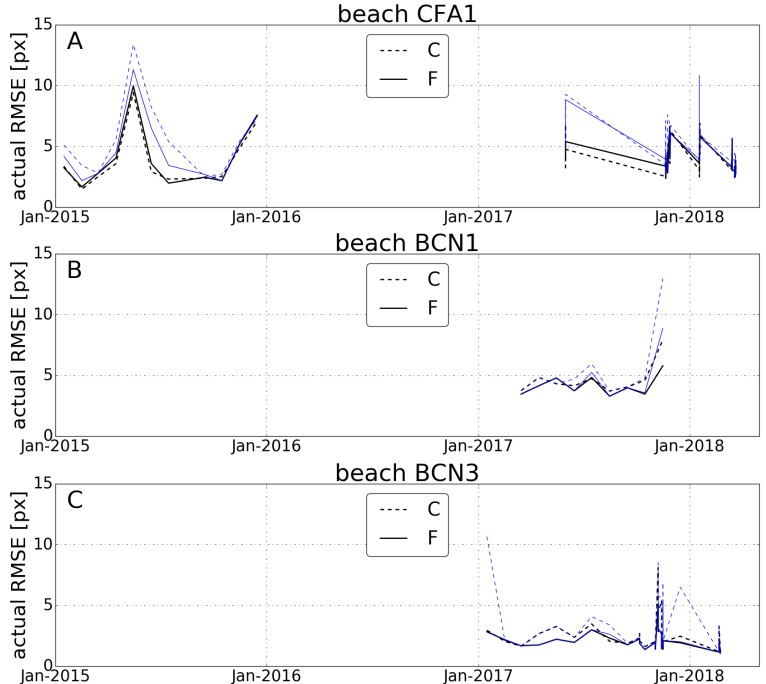

**Figure 11.** Actual RMSE in pixels (2 px = 1 m) of the automatic shorelines in the different dates of beaches (**A**) CFA1, (**B**) BCN1 and (**C**) BCN3, for the combined (dashed) and filtered (solid) shorelines, using a threshold of 20 px (black) or no threshold (blue) in the self-computed errors.

## 4. Discussion

### 4.1. Sensitivity Analysis

The default values of the model parameters (case 01 in Table 2) have been chosen for providing the smallest errors after varying them within wide ranges (not shown). A detailed sensitivity analysis to slight variations of these default values has also been done (cases 02–13 in Table 2). The corresponding results of the global RMSE, bias, and percentage of success (shown in Table 4 for a threshold of 20 px) show insignificant changes in most cases. This proves the robustness of the selected default values.

In particular, varying the parameters $\Delta\tau$, $a_t$, $\Delta t$, $a_s$ and $\Delta s$ (cases 02–05), related to the initial filtering of the raw shorelines in the combination step (Section 2.3.2), has a negligible effect on the final results. Also, the order of application of the time and space filters was changed and it had no influence either. This behaviour was expected (to some extent) given that the filtered raw shorelines are only used to evaluate the goodness of each of the original raw shorelines and to compute their weights for the subsequent combination process. Since the combined shorelines are then built with a weighted average of the unfiltered original raw shorelines (and not with the filtered raw shorelines), the method is robust to changes of these filtering parameters. Case 06 ($\alpha_1 = 1$ in Equation (1), related with the weight computation), give the same RMSE as the default case but worse bias and percentage of success. Thereby, using is $\alpha_1 = 2$ is clearly more appropriate, which is consistent with the classic approach of using the inverse of the variances as weights when combining multiple estimators [28]. Case 07 ($b_s = 2$, related with the computation of the error $E_{ij}^{c1}$) gives a slightly worse percentage of success so the default value is more recommendable.

Cases 08–13 are related with the parameters of the filtering process of the combined shorelines and, therefore, only affect the filtered shorelines. The role of $\alpha_2$ in Equation (6) is minor although using a value $\alpha_2 = 2$ works slightly better. Results slightly improve when increasing $m_t$ (i.e., using a longer window for the time filtering, cases 10–11) but the default value $m_t = 3$ points and the value $m_t = 5$ points are very similar. On the other hand, results are insensitive to increasing $m_s$ in the proposed range (i.e., using a larger window for the space filtering, cases 12–13). This is an interesting

result: in the filtering step, a time window of 1 h per side ($m_t = 3$ points) and a space window of 2 m per side ($m_s = 5$ points) are good enough. In particular, some interesting morphological patterns like beach cusps occur at length scales of some 10 m [34] and could be accurately retained by the automatic shorelines (see an example in Figure 3). If the goal was to get a gross approximation of the shoreline to obtain, e.g., the beach width, increasing significantly the value of $m_s$ in the filtering process (e.g., up to 21 points, so a 10 m window size) would be an option to obtain reliable filtered shorelines without gaps (the price being to miss the small features of the shoreline). In the same line, the shoreline time evolution linked to storm surges or astronomical tides (if present) can be captured by the methodology using the recommended value $m_t = 3$ points but using a larger value of this parameter would filter such fast variability providing the long-term trends.

**Table 4.** Sensitivity of the global actual RMSE and bias (in pixels), together with the percentage of success, of the combined shorelines and the filtered shorelines to the parameter values described in Table 2, using a threshold for the self-computed errors of 20 px and including all studied dates and beaches.

| Case | Combined | | | Filtered | | |
|---|---|---|---|---|---|---|
| | RMSE [m] | Bias [m] | Success [%] | RMSE [m] | Bias [m] | Success [%] |
| 01 | 1.7 | −0.03 | 96 | 1.7 | −0.02 | 97 |
| 02 | 1.7 | −0.04 | 96 | 1.7 | −0.04 | 97 |
| 03 | 1.8 | 0.01 | 96 | 1.7 | 0.06 | 97 |
| 04 | 1.8 | −0.03 | 96 | 1.7 | −0.02 | 97 |
| 05 | 1.7 | −0.01 | 96 | 1.7 | 0.00 | 97 |
| 06 | 1.8 | −0.09 | 86 | 1.8 | −0.05 | 89 |
| 07 | 1.7 | −0.01 | 92 | 1.7 | 0.00 | 94 |
| 08 | 1.7 | −0.03 | 96 | 1.9 | −0.04 | 96 |
| 09 | 1.7 | −0.03 | 96 | 1.7 | −0.01 | 98 |
| 10 | 1.7 | −0.03 | 96 | 1.8 | −0.03 | 96 |
| 11 | 1.7 | −0.03 | 96 | 1.7 | −0.01 | 98 |
| 12 | 1.7 | −0.03 | 96 | 1.7 | −0.02 | 97 |
| 13 | 1.7 | −0.03 | 96 | 1.7 | −0.01 | 97 |

A more sophisticated space-time filtering based on EOF (Empirical Orthogonal Functions) was initially proposed and included in the combination step. This alternative filtering procedure, which considered the dominant modes to reconstruct a smoothed signal (smoothed at once in space and time), was disregarded since it complicated the methodology without improving the final results.

*4.2. Interpretation of the Results*

The presented methodology provides several outcomes: two types of shorelines (combined and filtered) and the corresponding self-computed errors that can be used for selecting a subset of points by imposing a threshold. Thereby it is important to establish which is the recommended option to be used in view of the obtained results.

On the one hand, the filtered shorelines are clearly better than the combined shorelines when no threshold is applied (Table 3). On the other hand, it has been proven that the self-computed errors are better proxies of the actual errors in the combined shorelines than in the filtered shorelines (see Figure 9 and the corresponding $r$ values in Section 3.2). This is not surprising because the self-computed errors are mainly obtained during the combination step and are more representative for the combined shorelines. The errors of the filtered shoreline come from a weighted average of the errors of the combined shorelines, becoming less meaningful. This also explains why applying a threshold in the filtered shorelines has a smaller effect than in the case of the combined shorelines.

In spite of the above, our recommendation is still to use the filtered shorelines with a certain threshold. The final choice for the threshold, however, will depend on the goal of the specific application of the obtained shorelines and the next guidelines can be followed. Imposing no threshold implies having a continuous shoreline (with all points) but less reliable (mean beach errors up to 3 m). Using a threshold of 10 px gives more reliable results but with significant gaps. Based on the results, our recommendation would be imposing a threshold of 20 px since it provides an interesting intermediate situation: beach errors remain in general smaller than 2 m with a percentage of obtained points larger than 95% (Table 3, bold numbers).

As explained in the results section, the self-computed errors of beach BCN4 (which are small) are not representative of the actual errors because they are dominated by a bias of about −2 m (Table 3). The planviews of this beach systematically show four different "colours" near the shoreline: yellow in the dry beach, brown in the wet area shoreward from the shoreline, light gray in the foam seaward to the shoreline and dark blue in the water (Figure 12A). The automatic method positions the shoreline in the light gray area, located systematically around 4 m seaward of the color change manually selected by the expert users (compare Figure 12B,C). In practical applications, the bias could be subtracted from the shoreline position to obtain more reliable shorelines (with zero bias), or also before applying the threshold in the self-computed errors, if needed.

As shown in Figure 11A and described in Section 3.2, even after filtering and applying the recommended threshold of 20 px the errors can be larger than 3 m in a few planviews of CFA1 (5 out of 55). There are two reasons why it is more difficult to extract the shorelines in Castelldefels than in Barcelona beaches. The first one is that the cameras are in a much lower location (30 m versus 142 m) which implies that from November until February there is always sun glare in one of the cameras. Figure 3 shows an example where the method does a good job despite the presence of sun glare in the central camera, which happens in most situations with sun glare, but in a few analysed cases the method is mislead by the strong light variation within the planview. The second reason is that Castelldefels is an open natural barred beach that shows a much more complex geomorphological behaviour than the artificial Barcelona non-barred beaches, which are protected by numerous groins and breakwaters. In particular, the Castelldefels beach often shows the presence of shore-parallel bars, crescentic bars, transverse bars and megacusps, which complicates the shoreline detection. For example, this is the case of the error peaks on 15 May and 15 December 2015 in Figure 11A. In general, the method detects the less accurate points (due to the presence of either a complex morphology or sun glare) and assigns them larger self-computed errors, so that when the threshold is reduced to 10 px the overall planview error is reduced significantly in Castelldefels beaches. However, under some circumstances, there can be small-scale ridge-and-runnel systems (probably reminiscent of attached transverse bars) with the typical presence of narrow wet areas inside the dry beach (in the runnels) that mislead the presented methodology: the expert users always select the most seaward shoreline whilst the methodology (the raw methods underneath) sometimes selects the most shoreward shoreline without detecting its mistake. An example occurs in CFA1 on 15 December 2015 (see Figure 13), where the shoreline is wrongly detected even after applying the threshold of 10 px, giving a mean planview actual error of 3.7 m (the largest one obtained in the 138 planviews for this threshold). This points at the main limitation of the present methodology: it is not conceived for the case when several shorelines appear in one transect and this forbids its usage in sites where this occurs (e.g., with a sand spit).

The camera resolution also plays a role. Within beach CFA1, the areas further away from the camera position, which have less resolution, show systematically larger errors. Beach BCN3, which has the best resolution for being close to the cameras, shows the smallest errors (Table 3). This would also happen in BCN4 if there was not the aforementioned bias problem. Finally, RMSE between 2.5 and 3 m (higher than the average) occur in beaches BCN3, BCN4 and BCN5 in some planviews of July and August, due to the presence of numerous people at the shoreline and the dry beach. In many other planviews the method does a good job capturing the shoreline despite the noise produced by the

people, sometimes paying the price of a lower success percentage (see Figure 4, with a highly occupied beach typical of summer).

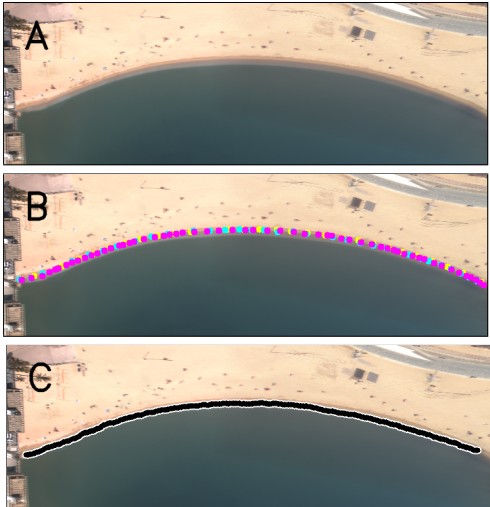

**Figure 12.** Examples of (**A**) original planview, (**B**) manual shorelines digitized by the three users (in different colors), (**C**) filtered shoreline (white and black) of beach BCN4 on 15 March 2017 at 12 h. The images do not have the original size of the planview but a zoom is applied to facilitate the visualization. In (**C**), the white points is the filtered shoreline, while the points of the filtered shoreline where $E^f < 10$ px are in black.

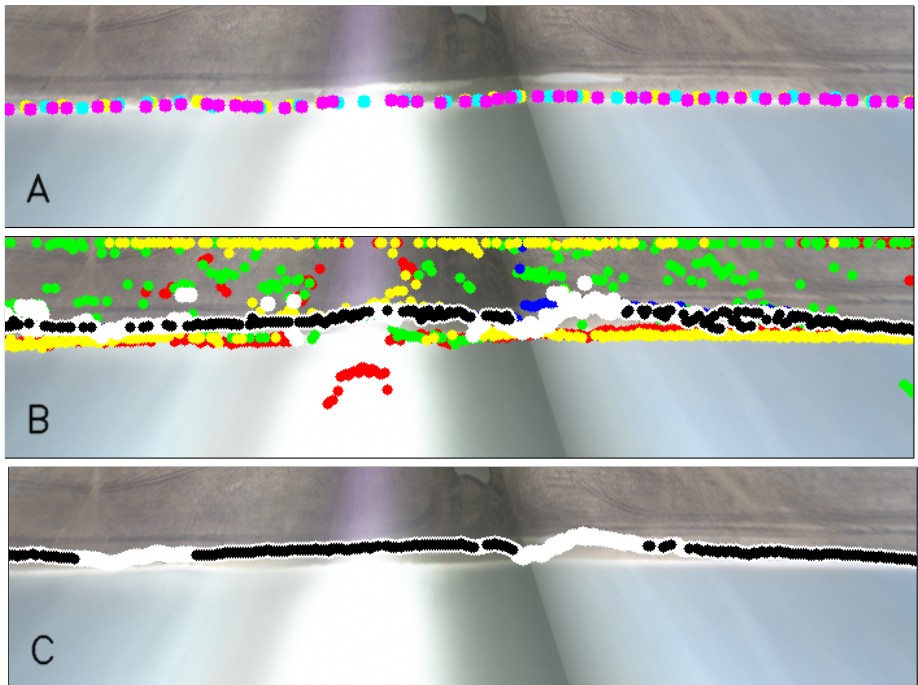

**Figure 13.** Equivalent to Figure 3, but of beach CFA1 on 15 December 2015 at 12 h. The images do not have the original size of the planview but a zoom is applied to facilitate the visualization. The presence of narrow wet areas in the dry beach misleads the methodology.

In any case, despite the great variety of analysed conditions, with and without the presence of wave breaking, complex sandbar morphologies, sun glare or high beach occupation, the obtained filtered shorelines are remarkably accurate in the great majority of planviews. One final remark: the outcome of our methodology can also be useful to initialize other shoreline detection methodologies. The continuous filtered shorelines obtained without threshold can be used as seeds or to find the initial

ROI (Region Of Interest), required by many shoreline extraction methodologies [22]. The accurate shorelines obtained with a small threshold can be used as training data in methods based on artificial neural networks or other machine learning techniques.

*4.3. Evaluation of the Raw Methods Used*

Once the filtered shoreline (and self-errors) are known, the goodness of the original raw methods can be assessed. This is a secondary outcome of the methodology. Table 5 shows the average (through all dates) of the RMSE of the raw shorelines as compared to the filtered ones, for each beach. The errors are considered only for the transects and dates where the self-error of the filtered shoreline is below a threshold of 20 px.

From Table 5, the errors are in general large because the raw shorelines are not filtered and include numerous outliers (Figures 3B, 4B, 5B and 13B). The channel S is the worst in CFA1, BCN1 and BCN2. On the other extreme, the ratio R/G clearly provides the best results in all the studied beaches (an example can be seen in Figure 6). The other three channels can also be good enough (different ones in the different beaches) and, most importantly, they may work in situations where R/G ratio fails.

**Table 5.** RMSE in meters of the shorelines obtained with the four initial raw methods (Hue, Saturation, Value and the ratio Red/Green) for each beach, including all studied dates.

| Beach | RMSE [m] | | | |
|-------|---|----|----|-----|
| | **H** | **S** | **V** | **R/G** |
| CFA1 | 14 | 31 | 15 | 12 |
| BCN1 | 12 | 78 | 54 | 7 |
| BCN2 | 38 | 53 | 16 | 4 |
| BCN3 | 46 | 14 | 31 | 10 |
| BCN4 | 16 | 15 | 14 | 9 |
| BCN5 | 8 | 15 | 25 | 10 |

The strength of the proposed methodology is to combine the different raw methods to obtain the best possible shoreline. However, if there is one raw method that has errors systematically larger than the others (e.g., the S channel in CFA1), the user could choose to suppress it and restart the methodology so as to slightly reduce the noise (although the methodology already gives small weights to noisy signals). In fact, the raw methods chosen in the present study are particularly simple in order to illustrate the interest and feasibility of the combination. As mentioned from the beginning, the methodology is open to incorporate other raw methods (including those applicable to radar or satellite images), and will automatically give higher weights to the best method. In fact, the errors in Table 5 are related to the weights given in the combination step, i.e., the smaller the error, the higher the weight.

## 5. Conclusions

A new fully-automatic methodology to extract the shoreline from video-images has been presented and applied to 138 planviews of six beaches in the region of Barcelona city (Western Mediterranean Sea). The ground truth is given by manually digitized shorelines, with inner errors of about 1 m. A weighted combination of four unfiltered raw shorelines obtained with simple methods has successfully allowed us to retrieve the shoreline with an error much smaller than those of the original signals—an application of *sensor fusion* to this problem. The assumption behind the combination is that the shoreline must be relatively smooth in space and time and the best weights for the combination turn out to be the inverse of a variance, which captures the (space and time) noise of the original raw signals. The combined shoreline can then be filtered in space and time and this further reduces the errors. The methodology also assigns a self-computed error to each retrieved point (in both the combined and the filtered shorelines), which is proved to be a good proxy for the actual error.

There are different approaches to apply the methodology: applying or not the space-time filter and imposing or not a threshold to the self-computed errors. Our results show that a good compromise is using a slightly filtered shoreline, which still captures the natural shoreline variability (at scales of some 10 m and a few hours), with a rather large threshold for the self-computed errors. With this configuration, the large RMSE of the initial raw shorelines (some 10–70 m) are reduced to less than 2.5 m in the great majority of analysed planviews (117 out of 138 planviews), with a percentage of retrieval success of 95% of the points. The global RMSE for all dates range from 1 to 2 m depending on the beach, with average biases smaller than 1 m. Only one of the six beaches shows a larger bias of 2.3 m, which increases the RMSE to 2.5 m. The obtained accuracy is fully satisfactory for many applications, despite the wide range of geomorphological conditions tested, including the presence of sun glare, storms and complex morphologies, as well as moments of high beach occupation. Moreover, the method automatically assigns larger self-computed errors to the points with larger actual errors, so that the final accuracy can be increased even further by diminishing the threshold for the self-computed errors, if needed (but reducing the amount of shoreline points recovered). The methodology is simple and versatile, allowing not only to use other shoreline detection methods as initial raw signals but also to apply it to other remote sensing techniques, such as satellites.

**Author Contributions:** The first two authors have equally contributed to the work. Conceptualization, G.S. and F.R.; methodology, G.S., F.R. and P.L.; software, G.S., F.R. and P.L.; validation, F.R., G.S. and J.A.; formal analysis, F.R., G.S., J.A. and P.L.; investigation, F.R., G.S., J.A. and P.L.; resources, F.R., G.S., J.A. and P.L.; writing—original draft preparation, F.R. and G.S.; writing—review and editing, F.R., G.S., J.A. and P.L.; visualization, G.S. and F.R.; supervision, F.R. and G.S.; project administration, F.R. and G.S.; funding acquisition, F.R. and G.S. All authors have read and agreed to the published version of the manuscript.

**Funding:** This research was funded by the Spanish Government (MINECO/FEDER) projects CTM2015-66225-C2-1-P, CTM2015-66225-C2-2-P, RTI2018-093941-B-C32 and RTI2018-093941-B-C33. The third author is funded by the project Cátedras-CONACyT 1146.

**Acknowledgments:** The authors are deeply grateful to J. Guillén, from the Institute of Marine Sciences (CSIC, Spain), for installing the first cameras in Barcelona two decades ago and for generously sharing his ideas and insights throughout all these years. The authors also acknowledge the useful suggestions from M. Soler, L. Auger and F. López.

**Conflicts of Interest:** The authors declare no conflict of interest.

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
