# Peer review of "Automatic Shoreline Detection from Video Images by Combining Information from Different Methods"

_remotesensing, doi:10.3390/rs12223717_

Round 1

Reviewer 1 Report

This is a good paper.

It would be great if the authors add some sentences to explain generation of the raw shorelines from a view point of physics of interacts of light and sand and water. In other words, give a brief explanation "water has always higher Hue values than the sand" (line 181) and "the water zone has systematically lower Saturation values" (line 182).

After show us the sound physics of the raw data, we can follow your philosophy, elsewise we have another philosophy, garbage in and garbage out. 

Author Response

Dear Reviewer,

We would like to thank this Reviewer for his/her positive comments on our manuscript. We have included his/her suggestions and those of the other Reviewer, and we think that the manuscript has improved significantly. Below, we answer the Reviewer’s comment, using underlined text (when needed) to highlight the specific changes made in the manuscript.

Yours sincerely,

Francesca Ribas, also on behalf of the other authors

-----------------------------------------------------------------

R1-Comment 1. It would be great if the authors add some sentences to explain generation of the raw shorelines from a viewpoint of physics of interactions of light and sand and water. In other words, give a brief explanation "water has always higher Hue values than the sand" (line 181) and "the water zone has systematically lower Saturation values" (line 182). After showing us the sound physics of the raw data, we can follow your philosophy, elsewise we have another philosophy, garbage in and garbage out.

R1-Reply 1. Following this Reviewer’s suggestion, in the first paragraph of Section 2.3.1 we have added a detailed explanation of the physics that justifies each method with the appropriate references (following also Comment 3 of the other Reviewer).

Regarding the comment “After showing us the sound physics of the raw data, we can follow your philosophy, elsewise we have another philosophy, garbage in and garbage out”, we emphasize that we have now explained the physics underneath the different raw methods, which justifies that they are often able to detect the shoreline. Further, the hypothesis of the proposed methodology (which is clearly written in the manuscript and is sustained by the results) is that the raw shorelines that are smooth both in space and time will correspond to the actual physical shoreline.

Reviewer 2 Report

The study combined different algorithms to develop a new automatic method for deriving the shorelines from the video stations. The overall effort is interesting and is meaningful to broad audience. Minor revision needs to be made before publication.

1) Figure 1. Add scale bar in left two images

2) Table 1. Please add the pixels sizes (in meter)

3) Line 173. Although they are well-known methods, it needs citations here to tell a broad audience.

4) Discussion is good to describe the outcomes of the current algorithm. But I am interesting to know how the actual physical and environmental conditions are affected the algorithms accuracy? Is your method can be applied to any type of the shoreline? What’s the limitation of the method? How the ratio of water vs land will affect the results? More discussion is appreciate.

Author Response

Dear Reviewer,

We would like to thank this Reviewer for his/her positive comments on our manuscript. We have included his/her suggestions and those of the other Reviewer, and we think that the manuscript has improved significantly. Below, we answer the Reviewer’s comments, using underlined text (when needed) to highlight the specific changes made in the manuscript.

Yours sincerely,

Francesca Ribas, also on behalf of the other authors

--------------------------------------------------------------------

R2-Comment 1. Figure 1. Add scale bar in left two images

R2-Reply 1. Following this Reviewer’s suggestion, we have added the missing scales.

R2-Comment 2. Table 1. Please add the pixels sizes (in meter)

R2-Reply 2. Following this Reviewer’s suggestion, we have added the pixels sizes in the caption of the table.

R2-Comment 3. Line 173. Although they are well-known methods, it needs citations here to tell a broad audience.

R2-Reply 3. This part of the manuscript has been extended, following also Comment 1 of the other Reviewer. Following this Reviewer’s suggestion, we have also introduced the references.

R2-Comment 4. Discussion is good to describe the outcomes of the current algorithm. But I am interested to know how the actual physical and environmental conditions affect the algorithms accuracy. Can your method be applied to any type of shoreline? What’s the limitation of the method? How the ratio of water vs land will affect the results? More discussion is appreciated.

R2-Reply 4. We first reply to the questions “Can your method be applied to any type of shoreline?” and “What’s the limitation of the method?”. In the last three paragraphs of Section 4.2 we already discuss the conditions under which our method has more difficulties to accurately detect the shoreline. These are mainly the presence of sun glare, the existence of complex geomorphological features and the presence of people on the beach. Notice that these conditions difficult the shoreline detection in any method, even including the manual digitization. In fact, as we also explain in this part of Section 4.2, the present methodology is able to detect the shoreline in most cases even under these complex conditions or, at least, identify the “wrong points” with larger self-computed errors.

The only exception is the situation where two shorelines are present because our method is not conceived for the case when several shorelines appear in one transect. In the present study this only occurs sporadically in one of the beaches (Castelldefels), linked to the presence of small-scale ridge-and-runnel systems with narrow wet areas inside the dry beach. However, this limitation would forbid the usage of this method in beaches with several shoreline positions in one transect (e.g., with a sand spit). We have added a sentence about this issue in Section 4.2 (line 486 of the manuscript with tracked changes). We do not see other limitations, as long as meaningful raw methods are used, of course. 

Regarding the question “How the ratio of water vs land will affect the results?”, we do not think this will have any effect on the results because the present methodology is based on detecting the shoreline along cross-shore transects (instead of being a global method that detects the shoreline in the whole image at once).

Round 2

Reviewer 1 Report

Good. No more comments.

Reviewer 2 Report

AUthors have revised the paper as suggested. Accept